# Specific Inhibition of HIF Activity: Can Peptides Lead the Way?

**DOI:** 10.3390/cancers13030410

**Published:** 2021-01-22

**Authors:** Ilias Mylonis, Georgia Chachami, George Simos

**Affiliations:** 1Laboratory of Biochemistry, Faculty of Medicine, University of Thessaly, 41500 Larissa, Greece; ghah@med.uth.gr; 2Gerald Bronfman Department of Oncology, Faculty of Medicine, McGill University, Montreal, QC H4A 3T2, Canada

**Keywords:** hypoxia, HIF, HIF-1α, HIF inhibition, peptide inhibitors

## Abstract

**Simple Summary:**

Cancer cells in solid tumors often experience lack of oxygen (hypoxia), which they overcome with the help of hypoxia inducible transcription factors (HIFs). When HIFs are activated, they stimulate the expression of many genes and cause the production of proteins that help cancer cells grow and migrate even in the presence of very little oxygen. Many experiments have shown that agents that block the activity of HIFs (HIF inhibitors) can prevent growth of cancer cells under hypoxia and, subsequently, hinder formation of malignant tumors or metastases. Most small chemical HIF inhibitors lack the selectivity required for development of safe anticancer drugs. On the other hand, peptides derived from HIFs themselves can be very selective HIF inhibitors by disrupting specific associations of HIFs with cellular components that are essential for HIF activation. This review discusses the nature of available peptide HIF inhibitors and their prospects as effective pharmaceuticals against cancer.

**Abstract:**

Reduced oxygen availability (hypoxia) is a characteristic of many disorders including cancer. Central components of the systemic and cellular response to hypoxia are the Hypoxia Inducible Factors (HIFs), a small family of heterodimeric transcription factors that directly or indirectly regulate the expression of hundreds of genes, the products of which mediate adaptive changes in processes that include metabolism, erythropoiesis, and angiogenesis. The overexpression of HIFs has been linked to the pathogenesis and progression of cancer. Moreover, evidence from cellular and animal models have convincingly shown that targeting HIFs represents a valid approach to treat hypoxia-related disorders. However, targeting transcription factors with small molecules is a very demanding task and development of HIF inhibitors with specificity and therapeutic potential has largely remained an unattainable challenge. Another promising approach to inhibit HIFs is to use peptides modelled after HIF subunit domains known to be involved in protein–protein interactions that are critical for HIF function. Introduction of these peptides into cells can inhibit, through competition, the activity of endogenous HIFs in a sequence and, therefore also isoform, specific manner. This review summarizes the involvement of HIFs in cancer and the approaches for targeting them, with a special focus on the development of peptide HIF inhibitors and their prospects as highly-specific pharmacological agents.

## 1. Introduction

Oxygen is indispensable for cellular metabolism, as it is essential for aerobic respiration and energy production. Moreover, oxygen has a vital role as a substrate in many enzymatic reactions that regulate diverse biological processes. Mammals use oxygen sensing and delivery mechanisms to match available oxygen to their tissue demands so as to fulfill their metabolic needs but also prevent toxicity caused by excess oxygen [1]. Lack of sufficient oxygen or hypoxia can arise from an imbalance between oxygen delivery and consumption, that can occur under either physiological or pathological conditions [2]. A frequent example is ischemia, in which infarctions impair oxygen delivery via the circulation and form hypoxic areas. Oxygen delivery is also impaired in solid tumors because of the irregular vascularization caused by an imbalance between pro and anti-angiogenic signals. In addition, the high oxygen consumption rates of the rapidly proliferating cancer cell intensify the formation of hypoxic niches within solid tumors [3].

At the cellular level, hypoxia triggers the stabilization of the hypoxia inducible factors (known as HIFs) [2]. HIFs are transcription factors that initiate a cascade of events such as metabolic reprogramming, induction of angiogenesis and erythropoiesis that in healthy tissues facilitate adaptation to low oxygen conditions. Especially in cancer, HIFs, in addition to upregulating genes involved in glucose and lipid metabolism or vascularization, can also promote cell proliferation, resistance to apoptosis, evasion of the immune response, genomic instability, invasion, and metastasis [4,5]. Considering the pivotal role of HIFs and hypoxia in cancer, it is of no surprise that HIFs have been long targeted as means of anti-cancer therapy [6,7].

Here, we review the different strategies that directly or indirectly target control points of the hypoxic signaling pathway and their application in disease models. We specifically highlight the use of peptides as effectors of HIF activity and discuss their future perspectives and clinical significance.

## 2. The HIF-Dependent Response to Hypoxia

### 2.1. The HIF Family

The HIF family of heterodimeric transcription factors consists of 3 HIFα members (namely HIF-1α, HIF-2α, and HIF-3α) and one HIFβ member (HIF-1β, best known as ARNT) [8]. HIF-1α is the most well studied member of the family and the first to be discovered by Semenza and coworkers [9,10,11] by its ability to bind to a hypoxia response element (HRE) in the 3′ enhancer of the human *EPO* gene. Unlike HIF-1α that can be expressed in all types of cells, HIF-2α, encoded by the EPAS1 gene, is expressed in a few tissues such as placenta, lungs, liver, and heart, and holds a central role in angiogenesis and erythropoiesis [12,13]. HIF-3α, the less studied isoform, has many spliced variants with distinct expression pattern [14] and diverging functionalities ranging from HIF-1 inhibition to transcriptional activation of HIF targets [15,16]. 

Detailed studies have shown that HIF-1α, as well as the other HIFα forms, are stabilized under hypoxia by an oxygen dependent mechanism (Figure 1) [17]. ARNT, which is constitutively expressed in cells regardless of oxygen concentration, associates with the HIFα subunits within the nucleus to form a functional heterodimer (HIF) that can bind to the HREs of hypoxia target genes and initiate the transcriptional hypoxic response [18]. Heterodimerization and DNA binding are mediated by the Per-Arnt-Sim (PAS) homology and basic helix-loop-helix (bHLH) domains, respectively, which are present at the N-terminal parts of all HIF subunits [18]. Structural data indicate that both HIF isoforms bind HRE sequences in an identical fashion. The α-helices of their bHLH domains associate with the major groove of the recognition motif and their PAS-A domains cooperate with the bHLH domains of the heterodimer to establish binding to DNA [19]. Furthermore, the PAS domains of both HIFα isoforms possess cavities, which can accommodate small ligands, but their size and distribution differs between isoforms [19,20].

The C-terminal parts of both HIF-1α and HIF-2α are also critical for function as they contain their oxygen dependent degradation domain (ODDD) as well as two distinct transactivation domains, N-TAD (overlapping with ODDD) and C-TAD (at the very C-terminus), which is responsible for the interaction between HIFα and the transcriptional coactivator proteins CBP/p300 (Figure 1) [21]. HIF-1 and HIF-2 activate a great number of genes (more than 1000) which can either be specific for each factor or common for both of them [6,8]. Domain-swapping experiments have suggested that HIF target gene specificity may be conferred by the N-TAD through its interaction with additional transcriptional co-regulators [22]. 

### 2.2. HIFs and Cellular Oxygen Sensing

The cellular oxygen sensing mechanism has been characterized in the previous decade mainly by the work of G. Semenza, Sir P. Ratcliffe and W. Kaelin (2019 Nobel prize in Physiology or Medicine). Their breakthrough experiments revealed that the HIFα subunits are subjected to hydroxylation in two specific prolyl residues when cells are grown under atmospheric oxygen concentrations (normoxia) [21]. This post-translational modification is essential for interaction with the Von Hippel-Lindau (pVHL) tumor suppressor protein that is part of a ubiquitin E3 ligase complex. As a result of this interaction, HIFα subunits are polyubiquitinated and targeted to the proteasome for destruction [23,24,25]. The enzymes catalyzing this hydroxylation are prolyl-hydroxylases PHD1, 2, and 3 in humans, also known as EGLN2,1 and 3. PHDs belong to the 2-oxoglutarate-dependent dioxygenase family and are thought to act as oxygen sensors since they use molecular oxygen as substrate. PHD2 is the most abundant and best studied isoform in cells [1,26,27,28]. A second oxygen dependent hydroxylation occurs at an asparagine residue in the C-TAD of HIFα and is catalyzed by a different oxygenase, called FIH (Factor Inhibiting HIF) [29,30]. FIH downregulates HIF transcriptional activity by impairing HIF binding to CBP/p300. According to all the above, HIFαs are constitutively produced but in the presence of normal oxygen concentrations both their expression and activity remain minimal. Under hypoxia, low O_2_ levels as well as the production of ROS by oxygen-starving mitochondria inhibit hydroxylation and HIFαs escape degradation, accumulate, and translocate inside the nucleus where they assemble with ARNT into transcriptionally active heterodimers (Figure 1) [31].

### 2.3. Oxygen-Independent Regulation of HIFs

Over the past few years, it has become clear that HIFs can also be regulated by mechanisms not directly affected by oxygen concentration. This regulation can occur at multiple levels including transcription, translation, post-translational modification, stabilization, nuclear translocation and activation of HIFαs [2,31,32,33]. Post-translational modifications of HIFαs probably hold the most important role in their regulation [34]. HIFαs are subjected to acetylation, s-nitrosylation and sumoylation, but the significance of these modifications for HIF activity is still a matter of debate. On the other hand, HIFα phosphorylation is much better characterized with clearly demonstrated importance in various cellular models. Both HIF-1α and HIF-2α can be directly phosphorylated by several kinases including GSK3, PLK3, ATM, PKA, CDK1, and the extent of their modification depends on cell type and extracellular signals [34,35]. Previous work from our lab has also shown that HIF-1α is a direct target of kinases ERK1/2 and CK1δ, modification by which has distinct outcomes on HIF-1 activity [36,37,38,39]. More specifically, while import of HIF-1α inside the nucleus is constitutive, mediated by the importin α/β as well as importins 4/7 [40,41], nuclear export of HIF-1α is regulated in an ERK1/2- and CRM1-dependent manner (Figure 1) [38,39]. Phosphorylation of HIF-1α by ERK1/2 at Ser641/643, which lay inside a small domain termed ETD (ERK Targeted Domain; amino acids 616–658), masks a nearby CRM1-dependent nuclear export signal (NES), thus inhibiting HIF-1α nuclear export and increasing HIF-1α nuclear concentration and HIF-1 transcriptional activity [38,39]. Lack of this phosphorylation allows CRM1 binding to HIF-1α and its subsequent translocation to the cytoplasm, where, interestingly, HIF-1α interacts with mortalin and takes part in the assembly of anti-apoptotic complex on the surface of the mitochondria [36,42]. This mode of HIF-1α regulation by ERK1/2 has been exploited for the development of peptide HIF-1 inhibitors modelled after the ETD amino acid sequence (see below). On the other hand, phosphorylation of HIF-1α by Ck1δ at Ser247 inside the PAS domain has a negative effect by inhibiting the ability of HIF-1α to associate with ARNT [37,38,39,43,44,45]. Regulation of the HIF-1 heterodimer assembly by phosphorylation as well as by MgcRacGAP [46,47] highlights the HIF-1α/ARNT interaction as a target for peptide inhibitors modelled after the PAS domain, an approach that has indeed been successfully tried (see also below). ERK1/2 and CK1δ also modify HIF-2α at distinct sites, and in this case, they both appear to regulate HIF-2 activity by affecting the distribution of HIF-2α between nucleus and cytoplasm [44,45]. 

In addition to direct phosphorylation, signaling pathways involving PI3K, ERK1/2 or p38 MAPK when activated by non-hypoxic stimuli such as growth factors (e.g., PDGF, TGF-β, IGF-1, and EGF), cytokines or hormones can also affect HIF activation by indirectly modulating its expression or stability [34,35,48,49,50,51,52,53,54]. As an example, heregulin (a member of the EGF family of growth factors) induces HIF-1 by activating the PI3K/Akt/mTOR pathway and increasing the rate of HIF-1α translation [55]. Other exemplary modes of HIF regulation include the involvement of ROS signaling (reviewed in [56]), transcription factors such as NF-kb [57] and STAT3 [58] that upregulate transcription of the gene encoding HIF-1α and many interacting proteins such as HSP90 and RACK1, which stabilize or destabilize HIF-1α, respectively [59,60,61]. HIFα stability is also affected by CO_2_ concentration as both in vivo and in vitro hypercapnia decrease HIFα protein levels independently of the PHD/pVHL-mediated degradation pathway and, most likely, via lysosomal proteolysis [62].

## 3. The Involvement of HIFs in Cancer

HIFs and especially HIF-1 influence several hallmarks of cancer such as genomic instability, tumor cell invasion, metastasis, and angiogenesis as well as suppression of the anti-tumor immune response [5,63]. Most importantly, HIF-1 holds a prominent role as mediator of the metabolic reprogramming that characterizes many types of cancer cells [8,64]. HIF-1 upregulates expression of most enzymes of glycolysis as well as expression of pyruvate dehydrogenase kinase 1 (PDK1). PDK1 phosphorylates and inactivates pyruvate dehydrogenase (PDH), which catalyzes conversion of pyruvate into acetyl-CoA. Thus, HIF-1 drives pyruvate, the product of glycolysis away from the TCA cycle and towards production of lactate, even in the presence of oxygen, a phenomenon known as Warburg effect [8,64]. Lipid metabolism is also influenced by a HIF-1 and several HIF-1 gene targets are involved in lipogenesis, which is generally favored in cancer via an increase in fatty acid uptake or synthesis and storage and simultaneous downregulation of fatty acid oxidation (reviewed in [4]). This often leads to accumulation of lipid droplets [37,65,66,67] and protects cancer cells from lipotoxicity [37,65,66]. HIF-dependent gene expression is also important for the adaptation and metabolism of cells surrounding a tumor (e.g., stromal cells), which are known to play an important role for cancer development [64,68].

There are numerous studies in which HIF-α proteins are found overexpressed in malignant tumors [5]. In principle, overexpression of HIFα isoforms is associated with poor clinical outcomes in patients with solid tumors [5,6]. Interestingly, HIF-1α was also found elevated and correlated with bad prognosis in hematological malignancies (reviewed in [69]). However, there are few reports indicating that HIF-1α overexpression may be connected to a positive outcome in certain cancer types including head and neck [70], non-small cell lung [71] and neuroblastoma [72]. 

Another feature that gives HIFs a special role in cancer progression is their ability to promote epithelial to mesenchymal transition (EMT) as well as resistance to chemo- or radio-therapy. HIFs facilitate EMT mainly by enhancing the expression of genes such as *TCF3*, *ZFHX1A/B*, and *TWIST,* which repress E-cadherin and epithelial type promoting factors, while, at the same time, the expression of mesenchymal type genes is increased [73,74]. Furthermore, HIF-mediated gene expression drives extracellular matrix remodeling, resistance to anoikis-related cell death and establishment of new cancer colonies, all of which facilitate the metastatic phenotype of hypoxic tumors [75]. Moreover, HIF-1 mediates chemoresistance by inducing expression of proteins that enhance drug efflux such as multidrug resistance 1 (MDR1) [76,77] and MRP2 [78] or anti-apoptotic proteins that promote drug resistance such as survivin [79,80]. HIF-1 is also implicated in resistance to radiation therapy since it counteracts the cytotoxic effects of radiation such as DNA damage and production of reactive oxygen species (ROS) [81,82]. 

Despite the unequivocal involvement of HIFs in the adaptation of cancer cells in the hypoxic tumor microenvironment, which promotes tumor progression, metastasis and resistance to therapy, it is still questionable whether HIFs by themselves are pro-oncogenic in a normal genetic background [83]. HIFs are active under physiological conditions such as embryonic development, immune system development, high-altitude adaptation and exercise and play an essential role in the maintenance of normal tissue homeostasis. HIF activation under these conditions does not trigger oncogenesis. Even when HIFα is constitutively activated due to pVHL function loss in renal cells, development of renal carcinoma requires additional mutations [84]. These issues have become especially important due the recent licensing and wide clinical administration of PHD inhibitors as HIF activators, and subsequent erythropoiesis inducers, for the treatment of patients suffering from renal anemia [85]. Long-term administration of these PHD inhibitors has not demonstrated tumor-initiating or tumor-promoting effects in either animal models or phase III clinical trials, possibly because competitive inhibition of PHD catalytic activity cannot cause permanent and irreversible HIF activation or pharmacological HIF induction is graded and cannot exceed a physiologically acceptable threshold. Nevertheless, as discussed below, the proof-of-principle for HIF inhibition as a valid anticancer strategy has been demonstrated in several cases and in both animal and human studies. 

## 4. HIF Inhibitors as Therapeutics

Activation of the HIF pathway is a common final aspect of many cancers arising from a variety of events, including intratumoral hypoxia, signaling molecules (reactive oxygen species, cytokines, and growth factors) and oncogenic transformation (e.g., *VHL* loss of function in renal carcinoma) [86]. As already discussed, HIF activation facilitates cancer progression and has been correlated with increased patient mortality [5]. Consequently, there has been great interest for the discovery of chemical agents and drugs that impair HIF activity and there is a fast growing list of such agents that act via various molecular mechanisms in cancer cells and tested for inhibition of tumor growth in animal models (comprehensibly reviewed in [7]). 

### 4.1. Chemical Agents as HIF Inhibitors

Transcription factors such as HIFs have been long considered undruggable because they do not contain a single targetable catalytic site (as enzymes do) but their activity relies on protein–DNA or protein–protein interactions, specific inhibition of which by small molecules is very challenging [87], especially when no detailed structural information is available. Nevertheless, numerous attempts have been made to identify HIF inhibiting agents, mainly by screening chemical libraries for their effect on cellular systems that facilitate detection of HIF transcriptional activity. The well-characterized agents discussed in this section (Table 1) are only used as examples in order to highlight the constituents and/or processes of the HIF activation pathway that when targeted lead to inhibition of HIF activity. In brief, different compounds have been shown to interfere with HIFα mRNA production, protein synthesis, stability, nuclear transport, heterodimerization, DNA binding, and transcriptional activity.

#### 4.1.1. Inhibition of HIFα mRNA Expression and Protein Synthesis

Although few in number, there are certain compounds that impair the transcription of *HIF1A* or *EPAS1* genes to mRNA. These include agents such as aminoflavone shown to affect both *HIF1A* transcription and translation in breast cancer cell lines and xenograft models [88], GL331 that interferes with the activation of *HIF1A* promoter in a lung cancer cell model [89] and anthracyclines (such as idarubicin) that inhibit both *HIF1A* and *EPAS1* transcription in pheochromocytoma cells and xenograft models [90]. A much larger group comprises compounds that decrease the translation rate of HIFα mRNA in several different ways. This group includes cardiac glycosides (e.g., digoxin) [91], topoisomerase inhibitors (e.g., topotecan) [92], agents that target components of the mTOR pathway (e.g., rapamycin) [109], steroids such as calcitriol that influence HIFα translation rates [93] or compounds that inhibit major signaling pathways such as sorafenib [94] or YC-1 [95]. This list also includes microtubule binding agents such as EF24 [96]. However, despite of the ability of this category of agents to interfere with the production of HIFα protein, most of these compounds do not specifically target the HIF-pathway nor can they differentially target HIF-1 or HIF-2, with few exceptions such as YC-1, which can specifically inhibit HIF-1 (but not HIF-2) expression in macrophages [110,111].

#### 4.1.2. Inhibition of HIFα Stability

A distinguishing feature of HIFαs is that they can be quickly destroyed in response to oxygen or other stimuli in proteasomal or lysosomal compartments [24,112]. So, there have been numerous studies investigating agents that can promote HIFα degradation even under hypoxia or tumor-related stabilizing conditions. Archetypical examples of this group are HSP90 inhibitors (e.g., 17-AAG) that lead to RACK1-dependent and VHL-independent ubiquitination of HIFα [60]. Another example are antioxidant agents (e.g., N-acetyl cysteine) that destabilize HIFα subunits in a PHD/VHL-mediated manner [97].

#### 4.1.3. Inhibition of HIF Heterodimerization

Agents that belong to this group target the PAS domains of HIF-1α and -2α thus interfering with HIFα heterodimerization with ARNT. For example, acriflavine (an antibacterial drug) can bind to the PAS-B domain of both HIFα subunits, which share significant homology in this part [98]. However, recent structural data of the HIFα PAS domains have shown that, despite the similarity between the HIF-1α and -2α PAS domains, the HIF-2α PAS-B contains a ligand-fitting cavity that is absent from the HIF-1α PAS-B [19,113]. Following these studies, subunit-specific inhibition of HIF-2α/ARNT dimerization was shown using a prototype chemical agent designed to bind to this cavity. More importantly, these compounds (PT2399, and PT2385) were efficient inhibitors of HIF-2 activity, exhibited anticancer properties in both cellular and animal renal carcinoma models and PT2385 administration to a patient with metastatic renal carcinoma (ccRCC) showed very promising results [99,114]. Furthermore, in a Phase I clinical trial with a population of patients with highly pretreated advanced ccRCC, PT2385 showed promising efficacy and favorable tolerability as a monotherapy, raising hopes for even better results in combination studies [115].

#### 4.1.4. Inhibition of HIFα Intranuclear Localization

Other chemicals such as PD98059 and U0126 or naturally occurring substances such as Kaempferol that inhibit the activation of ERK1/2 pathway, impair the phosphorylation of both HIFα subunits, and thus promote CRM1-dependent nuclear export of HIFα, subsequent impeding HIF transcriptional activity and proliferation of cancer cells [39,42,44,102].

#### 4.1.5. Inhibition of HIF DNA-Binding and Transcriptional Activity

DNA intercalating agents such as echinomycin or doxorubicin impair HIF-1 and HIF-2 binding to chromatin and, thus, inhibit HIF transcriptional activity [103,104]. On the other hand, chetomin targets the transcriptional activity of HIFs by influencing its ability to form a functional complex with transcriptional coactivators CBP/P300 [105]. 

Taken together, this growing number of preclinical data on chemical agents that inhibit HIF-pathway activation suggests that HIF inhibitors can improve the current treatment in many human cancer types. However, apart from the selective HIF-2 inhibitor PT2385 administered to patients with ccRCC in phase I trials and showing good tolerance [99,115], all other types of inhibitors discussed in this section represent either agents with tolerance limitations or chemicals that lack the selectivity for the hypoxic machinery or are not HIF-isoform specific. 

### 4.2. Peptides as HIF Modulators

An approach that may lead to selective and isoform-specific inhibition or activation of HIFs is the disruption of protein–protein or protein–DNA interactions, using peptides modelled after the amino acid sequences of the HIFα isoforms. These peptides, when delivered inside cells, may compete with the endogenous HIFαs for critical interactions, thereby inhibiting HIF activity or impair their association with inhibitory proteins, thereby stimulating HIF activity. Of course, development of such peptide agents requires very detailed mapping and functional characterization of the specific domains they are modeled after (Figure 2) as well as practical ways for their administration and intracellular or intranuclear delivery. In the case of HIFαs, the peptide inhibition/activation approach is facilitated by their modular structure, having well separated DNA-binding and Transactivation/Regulatory domains, which makes targeting individual domains and activation steps more feasible. Concerning delivery of peptides, they can be overexpressed in cells using transfection with suitable vectors, but this is not a practical method, especially when it is intended to be developed for therapeutic purposes. An emerging approach to deliver peptides that modulate protein–protein interactions as means of treatment relies on protein transduction technology. This method involves small amino acid sequences of various origins (listed in detail in [116]) that are able to cross cellular membranes and deliver associated cargo intracellularly. Among these cell permeable peptides, HIV-1 Transactivator of Transcription (TAT) peptide has been extensively studied and has been frequently used as means of internalizing biomolecules with therapeutic potential [117]. Concerning HIFs, protein transduction technology was initially used in order to target interactors of the ODDD of HIF-1α and enhance HIF activity as potential therapeutic approach in ischemic diseases. Peptides derived from both the N-terminal (residues 343–417) and C-terminal (residues 549–582) part of the HIF-1α ODD and fused to the TAT sequence could efficiently penetrate into cells, stabilize HIF-1α even under normoxia, by competing for its PHD-dependent hydroxylation, and promote angiogenesis in both in vitro and animal models [118]. Moreover, a smaller ODD peptide fused to TAT could induce HIF-1α and promote VEGF production in rat cortical neurons possibly by the same degradation-protective mechanism [119].

#### 4.2.1. Peptide Inhibition of HIF Heterodimerization

The feasibility of inhibiting HIF heterodimerization using peptides stems from early studies which demonstrated that shorter splice variants of HIF-3α (IPAS) are expressed in human and mouse tissues and downregulate HIF activity [15,120]. These polypeptides comprise either the bHLH, PAS, and N-TAD or the bHLH and PAS domains with a length of 667 and 307 amino acid residues (aa), respectively, and can negatively regulate HIFs by inhibiting HRE-mediated transcription [15,121]. The various HIF-3α forms act as dominant negative regulators by forming transcriptionally inactive complexes with HIF-1α or HIF-2α that prohibit effective interaction with ARNT [15,120]. Along the same line, when a HIF-1α-derived polypeptide (dnHIF-1α; aa 30-389) containing the bHLH and PAS domain was expressed in pancreatic adenocarcinoma cell lines it acted as a dominant negative mutant (apparently by forming an DNA-binding but inactive complex with ARNT) as it reduced binding of endogenous HIF-1 to its target promoters and, subsequently, inhibited its activity. The inhibitory effectiveness of dnHIF-1α was demonstrated by causing reduced glucose uptake and cancer cell growth in both pancreatic cancer cells and xenograft animal models [122]. ARNT interacting peptide (Ainp1) is a 59 aa long peptide of unknown function, the mRNA of which is expressed in many human tissues. Ainp1 interacts with the bHLH domain of ARNT. Transient introduction of Ainp1 into cells led to ARNT sequestration to the cytoplasm, lower HIF-1α/ARNT complex formation, and reduced HIF-1 activity in Hep3B cells [123]. Furthermore, TAT-fused Ainp1 could localize inside the cell nucleus and suppress HIF-1 activity in three different cell lines by interfering with the bHLH domain of ARNT (Table 1) [100]. 

Another cell permeable peptide that targets the HIF-1α/ARNT association is the cyclic hexapeptide cyclo-CLLFVY isolated from a 3.2 million plasmid library coding for hexapeptides [101]. The synthetic hexapeptide could efficiently bind to the PAS-B domain of HIF-1α (but not HIF-2α) in vitro. MCF-7 and U2OS cells cultured in the presence of this peptide fused to the TAT epitope exhibited lower HIF-1α/ARNT association and decreased HIF-1 transcriptional activity (Table 1). Following these results, the same group engineered human cells conditionally expressing this HIF-1 inhibitory cyclic peptide. Inhibition of HIF-1 dimerization and hypoxia-response signaling was also verified in these engineered cells. Moreover, inhibition of HIF-1 by the cyclic peptide sensitized cells to the glycolysis inhibitor 2-deoxyglucose leading to reduced cell viability, showing synthetic lethality upon both inhibition of HIF-1 and glycolysis in cells under hypoxia [124].

#### 4.2.2. Peptide Inhibition of HIF-Dependent Transactivation

Another target of peptide-mediated HIF inhibition is the HIFα/p300 binding interface. Initial studies used truncation mutants to map the minimum amino acid sequences that are required for the association between HIF-1α and p300. It was shown that 116 residues of p300 (aa 302–418) comprising its CH1 domain and 41 C-terminal HIF-1α residues (aa 786–826; C-TAD) were necessary for efficient HIF-1α/p300 binding [125]. Peptides that derive from HIF-1α C-TAD (aa 766–826 and aa 786–826) or p300 CH1 peptides reduced *EPO* promoter activity when overexpressed in HCT-116 colon and MDA-MB435 breast carcinoma cell lines [125]. Importantly, the expression of the HIF-1α C-TAD polypeptides could effectively suppress hypoxic signaling and tumor growth in a xenograft model. However, the same peptides also reduced STAT2 activity that requires the same CH1 domain of p300 for its transactivation, suggesting that binding of the HIF-1α peptides to p300 could affect the activity of other transcription factors in addition to HIF-1, although signaling pathways that relied on transcription factor binding with p300 domains other than CH1 were not impaired [125]. Recent studies have performed a more detailed structural investigation of the binding interface between the HIF-1α C-TAD region (aa 776–826) and the p300 CH1 domain (aa 330–420) [126]. NMR data revealed that the HIF-1α C-TAD region consists of three helical subdomains and wraps around the p300 CH1 domain with helices 2 and 3 being more important for HIF-1α/p300 association. Structural information from different HIF-1 C-TAD truncation and p300 point mutants revealed that helix 3 is essential for HIF-1α binding to a p300 binding patch. In this context, different peptide and phage display libraries as well as synthetic HIF-1α (aa 812–826)-derived peptides were examined for their binding to p300 and their adopted conformation when bound to the CH1 domain. Designed synthetic and constrained peptides that were able to adopt α-helical conformation in their bound state to p300 were more efficient competitors of HIF-1α C-TAD and may represent lead compounds for the development of peptidomimetic HIF-1α/p300 inhibitors (see also below) [126,127]. 

#### 4.2.3. Inhibition of HIF-1α Nuclear Accumulation and Nuclear Interactions

As already mentioned, phosphorylation of Ser641/643 in the ETD domain of HIF-1α by p42/44 MAPK (ERK1/2), which are often activated in human cancers [128], is essential for HIF-1α nuclear accumulation and activity because it masks a CRM1-dependent NES [38]. Initial studies from our group showed that overexpression of ETD, as a 43-amino acid long free peptide, competed with endogenous HIF-1α and inhibited its activity in HeLa and Huh7 cells [36]. The specificity of this inhibition was shown by the fact that a mutant form of the ETD that could not be phosphorylated by ERKs (ETD-SA form, in which Ser641/643 were converted into Ala) left HIF-1 activity unaffected. In contrast, other mutant variants of the ETD that either mimic its phosphorylation by ERK (ETD-SE form, in which Ser641 was converted into Glu) or are constitutively nuclear (ETD-IA form in which the NES was mutated) were as efficient as their wild-type form in inhibiting HIF-1 activity, suggesting that the ETD peptides could act in either or both of two ways: as competitors of HIF-1α phosphorylation, being themselves ERK substrates, and/or as competitors of a phosphorylation-dependent essential nuclear interaction of the endogenous HIF-1α ETD. A better analysis of the effects of these inhibitory peptides became possible when they were delivered directly to the cells as cell-penetrating TAT-fusion recombinant moieties (Table 1) [36]. In this way, they were shown to efficiently disable the ERK-dependent transcriptional activity of HIF-1 without interfering with HIF-1α stabilization or affecting HIF-2 activity, as the ETD sequence is unique in HIF-1α. Moreover, administration of the inhibitory TAT-ETD peptides resulted in severe phenotypic defects, exclusively under hypoxia. These defects included inability of the cancer cells to metabolically adapt, migrate or form single cell colonies under low oxygen conditions. Furthermore, even though these peptides did not affect cell viability under normoxia, they significantly increased cell death rate by inducing apoptosis under hypoxia. Overall, these results underlined the significance of ERK-dependent phosphorylation of HIF-1α for the transcriptional response to hypoxia as well as the prospect of TAT-ETD peptides or their peptidomimetics as highly selective and isoform-specific inhibitors of the HIF-1 pathway in cancer cells.

## 5. Perspectives–Conclusions

Increased levels of HIF activity are a common feature of many cancer types and cancer cells that depend on HIF activity may be vulnerable to HIF inhibition alone or in combination with other more traditional treatments. However, most of the chemical HIF inhibitors identified so far suffer from poor selectivity against HIFs or limited patient tolerance as they target a broad range of cellular functions and potentially also interfere with the function of healthy cells. HIF activation and regulation involves many protein–protein interactions (PPIs) [59], targeting of which may have more selective and less toxic effects. PPI binding interfaces are usually large, shallow, and hydrophobic, which makes them largely intractable to small chemicals that can normally target well-defined cavities of enzymes or receptors. In contrast, the larger size and flexible backbones of peptides make them capable of binding and blocking the larger grooves or clefts of interacting interfaces [129]. Therefore, peptides that mimic endogenous interacting partners can very selectively and potently inhibit their association. Along this line, carefully characterized peptide sequences that target specific PPIs essential for HIF activation steps can be much more effective than small molecule inhibitors. However, clinical application of peptides presents major limitations due to their poor oral bioavailability, low cell permeability, and short plasma half-life. It is, therefore, a challenge to create peptide inhibitors that can share, at least some of, the advantages of small chemical inhibitors such as manufacturing cost, shelf life, oral delivery, metabolic stability, pharmacokinetics, bio-distribution, and permeability [130]. 

Once peptide–protein recognition mechanisms have been deciphered and characterized, a promising strategy is to improve the pharmaceutical properties of peptides through stabilization, fusion with cell-penetrating moieties, or even conversion into non-natural scaffolds, creating, thus, novel compounds often referred to as peptidomimetics [116,131]. Stabilization of peptides and improvement of their pharmacokinetic properties in vivo may include C-terminal amidation or N-terminal acetylation, conjugation with lipids or pegylated side chains or covalent "stapling" of helical regions along the peptide backbone [132]. More advanced strategies include construction of macrocycles (molecules containing a twelve or more-membered ring) through peptide cyclization, introduction of non-natural amino acid analogs or enantiomer (D-)amino acids and creation of peptide-inspired foldamers (chain molecules or oligomers that mimic the ability of peptides to fold into well-defined conformations, such as helices and β-sheets) [133,134]. These peptidomimetics may have small molecule drug properties like long in vivo stability, while maintaining robust affinity, specificity and minimal toxicity.

Such an approach has already been tried in the case of HIFs with promising results. Based on the sequence of HIF-1α C-TAD helix 2 (that is important for HIF-1α/p300 interaction) a hydrogen bond surrogate (HBS) α-helical peptide was synthesized that had better affinity for p300 and was effective in down-regulating HIF-1 activity in HeLa cells almost as efficiently as chetomin but without its toxic effects (Table 1) [106]. In an HBS peptide, a carbon–carbon bond replaces the intramolecular hydrogen bond, further stabilizing the α-helical conformation and promoting resistance to proteolysis. The same research group subsequently produced an HBS mimic of HIF-1α C-TAD helix 3, which, in addition to inhibiting HIF-1 activity in cancer-cell-based assays (Table 1), also impaired tumor growth when administered to a mouse xenograft tumor model, exhibiting, thus, much better in vitro and in vivo efficacy than the respective unconstrained peptide [107]. Furthermore, an oxopiperazine helix mimetic (OHM) of HIF-1α C-TAD helix 3, in which nitrogen atoms of neighboring backbone amides in a tetrapeptide are constrained with ethylene bridges, was able, despite its small size, to inhibit both HIF-1 activity and tumor growth rate in vitro and in vivo, respectively [108]. These results suggest that certain PPIs can depend on few protein side chains, so-called “hot-spot” residues, and mimicking the relative positioning and disposition of these important residues on synthetic scaffolds may lead to the production of small molecules that combine the advantages of both chemical and peptide inhibitors.

In conclusion, peptides as modulators of PPIs involving HIFs are not only important for in vitro experimentation and the elucidation of regulatory mechanisms but can also lead the way for the design and development of novel peptide analogues or synthetic peptidomimetics that can target exclusively HIF-overexpressing cells with great selectivity, low toxicity and pharmacological properties similar to those of small molecule chemical inhibitors. The effectiveness and advantages of HIF peptide inhibitors summarized herein, certainly provide proof-of-principle for the development of novel therapeutic options based on the interruption of PPIs that are essential for the adaptation of cancer cells to the hypoxic tumor micro-environment and urge for better and high resolution structural characterization of the corresponding HIFα PPI domains.

## Figures and Tables

**Figure 1 cancers-13-00410-f001:**
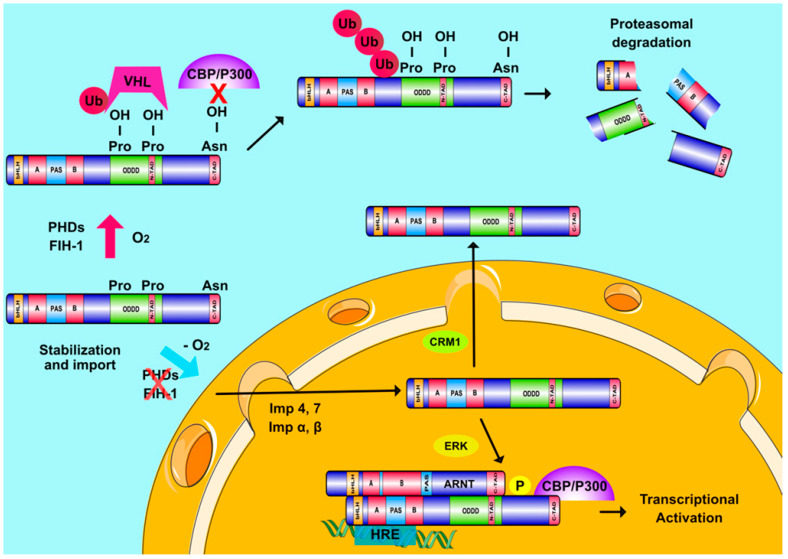
Regulation of HIFα subunits and Hypoxia Inducible Factors (HIF) transcriptional activity. When oxygen is abundant, PHDs and FIH hydroxylate two proline and one asparagine residue inside the ODDD and C-TAD regions of HIFα, respectively. Prolyl-hydroxylation leads to pVHL-mediated ubiquitination of HIFα subunits and their destruction in the proteasome while asparaginyl hydroxylation also inhibits HIFα interaction with CBP/P300. When oxygen levels drop, hydroxylases become inactive and HIFα subunits are stabilized and transported into the nucleus with the help of multiple importins (Imp α, β, 4, and 7). ERK-mediated phosphorylation of HIFα subunits ensures their nuclear accumulation by abrogating the association of HIFα with exportin CRM1. Nuclear HIFα subunits form a functional complex with ARNT, which binds to HREs on DNA and coactivators, such as CBP/p300 to induce transcription of hypoxia-regulated genes.

**Figure 2 cancers-13-00410-f002:**
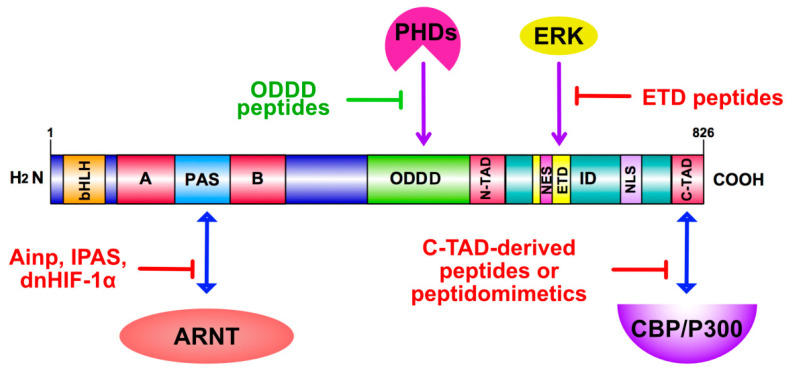
Peptide-mediated HIFα targeting. Peptides target the association of distinct HIF-1α domains (as indicated) with components of the hypoxia signaling or transcription machinery, such as ARNT, PHDs and CBP/P300 (activating peptides in green, inhibiting in red). Modification of HIF-1α by ERK and its phosphorylation-dependent nuclear interactions can also be inhibited by ERK Targeted Domain (ETD)-derived peptides.

**Table 1 cancers-13-00410-t001:** Chemical and peptide or peptidomimetic inhibitors of HIFs discussed in this article.

Inhibited Process	Nature of Inhibitor	Inhibitor	Active Concentr.*	Ref
HIFα synthesis	Chemical	Aminoflavone	0.25–0.5 μM	[88]
		GL331	10 μM	[89]
		Idarubicin	0.625 μM	[90]
		Digoxin	0.1 μM	[91]
		Topotecan	0.05–0.1 μM	[92]
		Calcitriol	0.1 μM	[93]
		Sorafenib	10 μM	[94]
		YC1	10–25 μM	[95]
		EF24	1 μM	[96]
HIFα stability	Chemical	17-AAG	0.5 μM	[60]
		NAC	10 mM	[97]
HIFα binding to ARNT	Chemical	Acriflavine	1–5 μM	[98]
		PT2399	2 μM	[99]
	Peptide	TAT-Ainp1	1–2 μM	[100]
		TAT-cyclo-CLLFVY	10–50 μM	[101]
HIFα nuclear accumulation and activity	Chemical	PD98057	50 μM	[39]
		U0126	5 μM	[42,44]
		Kaempferol	5–10 μM	[102]
		Echinomycin	1–5 nM	[103]
		Doxorubicin	0.2–1 μM	[104]
		Chetomin	10 nM	[105]
	Peptide	TAT-EDT	0.4 μM	[36]
	Peptido-mimetic	HBS 2 (C-TAD helix 2 mimic)	1 μM	[106]
		HBS 1 (C-TAD helix 3 mimic)	50 μM	[107]
		OHM 1 (C-TAD helix 3 mimic)	10 μM	[108]

* Concentration of inhibitor that causes significant reduction (close to or more than 50%) in HIFα expression and/or HIF transcriptional activity when administered to HIF-expressing cultured cells.

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
