# Peer review of "Specific Inhibition of HIF Activity: Can Peptides Lead the Way?"

_cancers, 2021, doi:10.3390/cancers13030410_

Round 1
Reviewer 1 Report
This review summarized the role of HIFs in cancers. The authors reviewed HIF-1 inhibitors with a special focus on peptide inhibitors. That is of great significance for the understanding of currently developed specific and effective HIF-1 inhibitors of macromolecules. Therefore, I suggest to publish in Cancers just after a minor typing revision according to the requirements of this Journal.
Author Response
Reviewer #1
This review summarized the role of HIFs in cancers. The authors reviewed HIF-1 inhibitors with a special focus on peptide inhibitors. That is of great significance for the understanding of currently developed specific and effective HIF-1 inhibitors of macromolecules. Therefore, I suggest to publish in Cancers just after a minor typing revision according to the requirements of this Journal.
Our response:
We thank the reviewer for appreciating our manuscript.
We have tried to correct all typing according to the requirements of Cancers.
Reviewer 2 Report
In this study, Mylonis et al present a solid piece of work that investigates the therapeutic potential for modulating Hypoxia Inducible Factor function specifically in cancers. They highlight the key mechanisms for HIF inhibition that have been developed with a specific focus on small peptides. This is an interesting and timely piece of work given the recent development of HI-disrupting small molecule compounds. There are a few key points to improve the overall manuscript.
- It is worth citing the data that shows the HIF TAD domains are responsible for mediating HIF-target gene specificity and HIF functional divergence through interactions with other transcriptional co factors in section 2.1. Cheng Jun-Hu et al., 2007 MBoC.
- The authors should include data from larger clinical trials showing efficacy of PT2385 compound in heavily pretreated patients with metastatic renal cell cancer in the “Inhibition of HIF Heterodimerization” section. Courney et al., JCO 2017
- The authors provide an excellent summary on the data in the literature. However, in the conclusion the authors should expand more and speculate upon the future of HIF peptide inhibitors development? What are the are the remaining critical questions the field must address?
- The most promising therapeutics targeting HIF activity are small molecules, such as PT 2385, which disrupt HIF2/ARNT interactions which have shown significant efficacy in clinical trials and have been pretty well tolerated. In the conclusion section the authors should expand upon the advantages/disadvantages of using peptide molecules versus small molecules for inhibition of HIF activity in cancer.
Author Response
Reviewer #2 (1)
In this study, Mylonis et al present a solid piece of work that investigates the therapeutic potential for modulating Hypoxia Inducible Factor function specifically in cancers. They highlight the key mechanisms for HIF inhibition that have been developed with a specific focus on small peptides. This is an interesting and timely piece of work given the recent development of HI-disrupting small molecule compounds. There are a few key points to improve the overall manuscript.
Our response:
We thank the reviewer for appreciating our work and for his/her constructive and fair comments, all of which we addressed (see below).
Reviewer #2 (2)
It is worth citing the data that shows the HIF TAD domains are responsible for mediating HIF-target gene specificity and HIF functional divergence through interactions with other transcriptional co factors in section 2.1. Cheng Jun-Hu et al., 2007 MBoC.
Our response:
A sentence has been added discussing the role of the N-TAD in target gene specificity with the corresponding reference in lines 79-81.
Reviewer #2 (3)
The authors should include data from larger clinical trials showing efficacy of PT2385 compound in heavily pretreated patients with metastatic renal cell cancer in the “Inhibition of HIF Heterodimerization” section. Courney et al., JCO 2017
Our response:
A sentence has been added discussing data from a larger clinical trial with the corresponding reference in lines 284-286.
Reviewer #2 (4)
The authors provide an excellent summary on the data in the literature. However, in the conclusion the authors should expand more and speculate upon the future of HIF peptide inhibitors development? What are the are the remaining critical questions the field must address?
Our response:
The last section (Perspectives-Conclusions) has been completely re-written and several new references have been added in order to expand on the future of HIF peptide inhibitor development by discussing peptidomimetics as an answer to critical questions concerning clinical application (Lines 440-499).
Reviewer #2 (5)
The most promising therapeutics targeting HIF activity are small molecules, such as PT 2385, which disrupt HIF2/ARNT interactions which have shown significant efficacy in clinical trials and have been pretty well tolerated. In the conclusion section the authors should expand upon the advantages/disadvantages of using peptide molecules versus small molecules for inhibition of HIF activity in cancer.
Our response:
As already mentioned above, the last section (Perspectives-Conclusions) has been revised and it now includes extensive discussion about advantages/disadvantages of peptide inhibitors versus small chemical inhibitors (Lines 443-457).
Reviewer 3 Report
General remark:
In this very well-written review article Mylonis and colleagues give a concise overview on how targeting of the hypoxia-inducible factor (HIF) signaling pathway could potentially improve cancer treatment. The authors correctly highlight that the expression of HIFs and numerous subsequent downstream signaling targets are increased in several primary tumors.
The authors correctly point out that specific targeting of HIF-isoforms by peptides might indeed be promising in cancer treatment. However, some of their statements concerning commercially available pharmacologic HIF inhibitors and their isoform-specific HIF inhibition should be refined as outlines below. Addressing these major concerns before acceptance for publication should improve the impact of the present review article and increase its readership.
Major concerns:
- “2.3 Oxygen-independent regulation of HIFs”: The authors expertly mention that HIF signaling is not only regulated by oxygen, but also by other factor, such as cell stress and inflammation. Recent evidence likewise indicates that high levels of carbon dioxide, which occur during ischemia and tumor hypoxia, reduce HIF-signaling activity in a pH-dependent manner (Selfridge et al., J Biol Chem2016, PMID: 27044749). Please amend and discuss the role of carbon dioxide as part of the adaptive response to hypoxia in 2-3 sentences.
- “3. The involvement of HIFs in cancer”: Whether increased HIF activity is required for tumor progression per seor simply a surrogate parameter for malignancy (and growing tumors) is, however, still unknown and thus topic of constant debate. So far clinical trials, investigating the role of pharmacologic upregulation of the HIF pathway by so called HIF Prolyl hydroxylase inhibitors to treat, for example, patients with anemia due to chronic kidney disease, have not shown induction of tumor frequencies in patients. Preclinical data likewise show conflicting results regarding the role of HIF during tumor progression (Sangahni and Haase, Adv Chronic Kidney Dis2019). Please expand this paragraph and discuss the role of HIF and tumor progression in more detail (include, for example, Cummins et al., Physiol Rev2020, PMID: 31539306).
- “Inhibition of HIFa mRNA expression and protein synthesis”: The authors correctly point out that many pharmacologic HIF inhibitors are not isoform-specific. However, recent preclinical data suggest that YC-1 in fact specifically inhibits HIF-1 (and not HIF-2) expression during LPS- and hypoxia-inflicted HIF signaling in macrophages (Strowitzki et al., Sci Rep2017, PMID: 29030625 and 31376431). Thus, please refine your statement and discuss above-mentioned articles.
Minor concerns:
Some stylistic inconsistencies could be corrected throughout the whole manuscript. For example: What do the authors mean by “infractions” (page 1, line 34)?
Author Response
Reviewer #3
General remark:
In this very well-written review article Mylonis and colleagues give a concise overview on how targeting of the hypoxia-inducible factor (HIF) signaling pathway could potentially improve cancer treatment. The authors correctly highlight that the expression of HIFs and numerous subsequent downstream signaling targets are increased in several primary tumors.
The authors correctly point out that specific targeting of HIF-isoforms by peptides might indeed be promising in cancer treatment. However, some of their statements concerning commercially available pharmacologic HIF inhibitors and their isoform-specific HIF inhibition should be refined as outlines below. Addressing these major concerns before acceptance for publication should improve the impact of the present review article and increase its readership.
Our response:
We thank the reviewer for appreciating our work and for his/her constructive and fair comments, all of which we addressed (see below).
Reviewer #3 (1)
Major concerns:
- “2.3 Oxygen-independent regulation of HIFs”: The authors expertly mention that HIF signaling is not only regulated by oxygen, but also by other factor, such as cell stress and inflammation. Recent evidence likewise indicates that high levels of carbon dioxide, which occur during ischemia and tumor hypoxia, reduce HIF-signaling activity in a pH-dependent manner (Selfridge et al., J Biol Chem2016, PMID: 27044749). Please amend and discuss the role of carbon dioxide as part of the adaptive response to hypoxia in 2-3 sentences.
Our response:
Text has been added in lines 157-160 discussing the involvement of CO2 in HIF regulation together with the relevant reference.
Reviewer #3 (2)
- “3. The involvement of HIFs in cancer”: Whether increased HIF activity is required for tumor progression per se or simply a surrogate parameter for malignancy (and growing tumors) is, however, still unknown and thus topic of constant debate. So far clinical trials, investigating the role of pharmacologic upregulation of the HIF pathway by so called HIF Prolyl hydroxylase inhibitors to treat, for example, patients with anemia due to chronic kidney disease, have not shown induction of tumor frequencies in patients. Preclinical data likewise show conflicting results regarding the role of HIF during tumor progression (Sangahni and Haase, Adv Chronic Kidney Dis2019). Please expand this paragraph and discuss the role of HIF and tumor progression in more detail (include, for example, Cummins et al., Physiol Rev2020, PMID: 31539306).
Our response:
A new paragraph has been added (lines 205-221) discussing the issues raised by the reviewer and the suggested references.
Reviewer #3 (3)
3.“Inhibition of HIFa mRNA expression and protein synthesis”: The authors correctly point out that many pharmacologic HIF inhibitors are not isoform-specific. However, recent preclinical data suggest that YC-1 in fact specifically inhibits HIF-1 (and not HIF-2) expression during LPS- and hypoxia-inflicted HIF signaling in macrophages (Strowitzki et al., Sci Rep2017, PMID: 29030625 and 31376431). Thus, please refine your statement and discuss above-mentioned articles.
Our response:
A sentence has been added clarifying the isoform- specificity of YC-1 and citing the requested articles in lines 264-265.
Reviewer #3 (4)
Some stylistic inconsistencies could be corrected throughout the whole manuscript. For example: What do the authors mean by “infractions” (page 1, line 34)?
Our response:
We have tried to correct all stylistic inconsistencies throughout the manuscript.
“Infractions” has been corrected to “infarctions”, we apologize for the typing mistake.
Reviewer 4 Report
In this manuscript, the authors review current strategies for developing HIF inhibitors and focus on summarizing the design and evaluation of peptide-based modulators. Such efforts could provide guidance for future work on this/related subject. I support the publication after the authors address the following minor issues:
For discussing the stabilization mechanism of HIF-1alpha (line 61- 72), the figure 1 should also be referenced with proper modifications on the original figure. Graphic interpretations are easier for readers to digest.
The author may want to include some brief discussions/perspectives in terms of the comparison between small molecule inhibitors and peptide-based inhibitors.
For discussing the potency and selectivity of representative inhibitors, it’s very important to bring in some quantitative data such as IC50 values for supporting the statement.
The author may want to check the references throughout the draft. For example, for statement like line 40: “HIFs are transcription factors that initiate a cascade of events....”, according citations should be included.
Author Response
Reviewer #4
In this manuscript, the authors review current strategies for developing HIF inhibitors and focus on summarizing the design and evaluation of peptide-based modulators. Such efforts could provide guidance for future work on this/related subject. I support the publication after the authors address the following minor issues:
Our response:
We thank the reviewer for appreciating our work and for his/her constructive and fair comments, all of which we addressed (see below).
Reviewer #4 (1)
For discussing the stabilization mechanism of HIF-1alpha (line 61- 72), the figure 1 should also be referenced with proper modifications on the original figure. Graphic interpretations are easier for readers to digest.
Our response:
We are thankful for this suggestion. Original Figure 1 has been modified to show the domain organization of HIFs and is also referenced in lines 63-82.
Reviewer #4 (2)
The author may want to include some brief discussions/perspectives in terms of the comparison between small molecule inhibitors and peptide-based inhibitors.
Our response:
As already mentioned, the last section (Perspectives-Conclusions) has been extensively revised and it now includes discussions/perspectives in terms of the comparison between small molecule inhibitors and peptide-based inhibitors(Lines 443-457).
Reviewer #4 (3)
For discussing the potency and selectivity of representative inhibitors, it’s very important to bring in some quantitative data such as IC50 values for supporting the statement.
Our response:
Unfortunately, most cited articles do not include determination of IC50 values for inhibitors, basically because inhibitors are mostly tested for their effects on HIF activity in cell-based biological assays by measuring either expression of a reporter gene or expression of endogenous HIF target genes. In these types of assays determination of IC50 values is not straight-forward because of solubility, permeability and toxicity issues. However, in order to indeed offer an idea for the potency of the discussed inhibitors, we have added a table which shows the concentration at which the inhibitors were reported to have significant effects in the aforementioned assays (Table 1).
Reviewer #4 (4)
The author may want to check the references throughout the draft. For example, for statement like line 40: “HIFs are transcription factors that initiate a cascade of events....”, according citations should be included.
Our response:
We have also re-checked the references throughout the text and tried to better integrate the references in the corresponding paragraphs.